# NetEvolve: Social Network Forecasting using Multi-Agent Reinforcement Learning with Interpretable Features

## ABSTRACT

Predicting how social networks change in the future is important in many applications. Results in social network research have shown that the change in the network can be explained by a small number of concepts, such as "homophily" and "transitivity". However, existing prediction methods require many latent features that are not connected to such concepts, making the methods' black boxes and their prediction results difficult to interpret, making them harder to derive scientific knowledge about social networks. In this study, we propose NetEvolve a novel multi-agent reinforcement learning-based method that predicts changes in a given social network. Given a sequence of changes as training data, NetEvolve learns the characteristics of the nodes with interpretable features, such as how the node feels rewards for connecting with similar people and the cost of the connection itself. Based on the learned feature, NetEvolve makes a forecast based on multi-agent simulation. NetEvolve achieves comparable or better accuracy than existing methods in predicting network changes in real-world social networks while keeping the prediction results interpretable.

## CCS CONCEPTS

• **Information systems** → **Data mining**; • **Applied computing** → **Sociology**; • **Computing methodologies** → **Multi-agent systems**.

## KEYWORDS

Network science, Time-series, Multi-agent system, Reinforcement learning

**ACM Reference Format:**
Anonymous Author(s). 2023. NetEvolve: Social Network Forecasting using Multi-Agent Reinforcement Learning with Interpretable Features. In *Proceedings of ACM Conference (Conference'17)*. ACM, New York, NY, USA, 10 pages. https://doi.org/10.1145/nnnnnnn.nnnnnnn

## 1 INTRODUCTION

A social network is a graph in which people are connected based on some relationship. Typical examples are the follower/followee relationships on Twitter ($\mathbb{X}$), the friend on Facebook, and collaborative relationships between researchers. In many cases, each social network node has attribute values that express its own characteristics. In the case of a researcher's co-authorship network, the

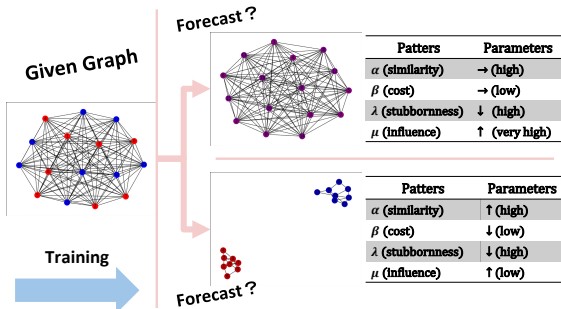

**Figure 1: NetEvolve forecasts the future network with interpretable features in real-world social networks. We define the features based on network science studies to simulate the phenomenon in real-world social networks such as homophily, heterophily, homogenization, and polarization [26, 33, 34, 39] which leads to the interpretation of the behavior of the nodes in the network.**

words contained in the researcher's published papers are examples. Social networks often change their structures and node attribute values over time. For example, the interests and opinions of each node change over time, and they change friendships in the network according to the change in their interests and opinions [22, 24, 39].

In this study, we aim to predict the changes in such dynamically changing social networks. The importance of social network prediction is growing, and the predicted interests and connections of a group of people are used for market size prediction [5], marketing [40], and analyzing opinion dynamics [30]. Thus, future prediction in social networks, such as predicting opinions and connections in social networks, is gaining importance and is expected to be used for prediction-based decision-making in social media in the future [3].

Methods for predicting dynamically changing social networks have been actively studied in recent years. Previous studies [11, 20, 23, 42] have used latent feature-based models that take into account interest propagation and utilize graph neural networks to predict how future connections in social networks will change. However, existing methods require a large number of latent features for prediction, making them black-box methods, which makes it hard to derive scientific knowledge from the model. Moreover, they do not fully consider theories considered in existing network research, such as transitivity [45].

From this background, we propose a novel method NetEvolve for predicting the future of social networks with interpretable features based on multi-agent reinforcement learning. Our study aims to forecast the change in their edges and attributes (interests or opinions) over time. In our study, we assume each node represents an agent in a reinforcement learning setting, and each node represents a rational agent, i.e., each node always moves to achieve a

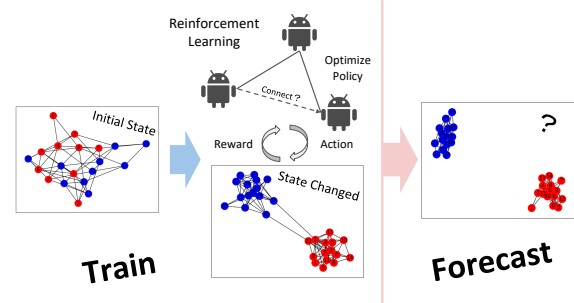

**Figure 2: Framework of NETEVOLVE** The proposed method defines a reward function and a policy function and learns the parameters from observed network sequences. The learned functions generate future network sequences by letting the agent behave as a network generator in a reinforcement learning environment.

higher reward in the network. For the reward and the policy functions, we designed the interpretable features based on knowledge of network science. Our research questions are two-fold;

**(RQ1)** *How to design the explainable reward and policy functions that can simulate the phenomena of social networks in a multi-agent reinforcement learning setting?*

**(RQ2)** *How well does the devised framework behave in terms of forecasting the real-world data?*

To answer (RQ1), we formulated a dynamic social network as an environment of multi-agent reinforcement learning, in which a node represents an agent in the environment. We assume the actions of nodes are to make/delete the edge and change their attribute. NETEVOLVE consists of three parts of processes (Figure 2);

**Step1.** Learning the reward functions of each node indicates in what situation the nodes feel comfortable.

**Step2.** Learning the policy of each node to learn the strategy to achieve a higher reward.

**Step3.** Based on the learned policy of each node, forecasts the future representation of the network based on the policy of each node.

We constructed the reward and policy functions based on a network science-based scheme. For the reward function, we designed a reward function as a linear combination of the similarity of connecting nodes' attributes and the cost function for remaining edges; this is based on the homophily effect [34] in the network, which means similar nodes have more connections and different attributes are difficult to connect to other nodes [14]. We designed a policy function that can illustrate various stories for getting high rewards.

The contributions of this study are the following. We proposed a novel method for forecasting social networks using interpretable features of network science and psychology knowledge based on multi-agent reinforcement learning. The number of parameters is smaller than that of existing methods, which improves the model's explanatory power and illustrates the characters of each node. Experiments using synthetic data show NETEVOLVE can simulate the phenomena in real-world social networks, such as homophily, heterophily, homogenization, and polarization[26, 33, 34, 39]. Moreover, to answer (RQ2), we conducted experiments using real-world

network data. The results show that by fitting the model to the real-world network data, NETEVOLVE forecasts the future network structure and the attribute more accurately than the previous works, which indicates NETEVOLVE well fits the real-world social phenomenon. The main advantages of NETEVOLVE are the following.

(1) **Interpretable**: Interpretability is improved by constructing a network science and psychology-based model.

(2) **Extensible**: can be easily extended to incorporate other network science and psychology known phenomena.

(3) **Effective**: Experimental results on real data show that our method outperforms existing methods in predicting edges in unobserved networks by 8% in accuracy.

**Reproducability:** Our code and datasets will be open-sourced on GitHub https://github.com/crowd4u/netevolve

## 2 RELATED WORKS

In this section, we describe the related works and discuss the differences between our work.

**Representation Learning for Social Networks** Representation learning for network data is the method for learning vectors encoding the network structure. In particular, several methods have been proposed within the framework of latent vector models based on probabilistic models [1], network embeddings [17, 38, 41, 44, 50], and graph neural networks [28, 29, 43]. In recent years, methods have been proposed to embed both node and attribute and track their changes [31], a method for tracking user interest in Twitter by embedding nodes and words in a dynamic network [50]. These methods were developed for acquiring graph and node features in observed networks and did not examine the prediction of networks at unobserved times.

**Reinforcement Learning** Multi-agent reinforcement learning is a framework in which multiple agents interact to learn behaviors that maximize their own or the group's satisfaction. Multi-agent reinforcement learning includes both fully cooperative and competitive tasks [10, 25]. Previous research [12] has focused on learning the optimal behavior of agents in an environment called a Markov game. Another study [48] defines social capital, which is the benefits society provides to individuals, in a game-based framework and predicts the emergence of new social network structures in a multi-agent reinforcement learning framework. Recent studies have attempted to conduct reinforcement learning for several graph mining tasks, such as representation learning, relational reasoning, and link prediction [32, 35].

While these studies have attempted to predict the optimal behavior of agents in a given situation and the associated emergence of network structures, they have not performed the task of generating unobserved time series networks from observed networks.

**Network Predictions** Recently, methods for predicting the social network were proposed. GraphSAGE [21] improves on GCN [28] by sampling from neighboring nodes during training to increase accuracy and speed, have been proposed as methods for generating new node features from the network structure. LFP [23] is a method for predicting changes in the latent variables of a node, taking into account their propagation in the network structure. CoNN [18] is a method for predicting changes in the opinions of a set of nodes as a crowd, and ELSM [19] is a method for augmenting the network

| Method / Property | LFP [23] | TensorCast [4] | Dyngraph2vec [16] | DualCast [27] | SINN [37] | NetEvolve |
|---|---|---|---|---|---|---|
| Network-forecast | ✔ | | ✔ | ✔ | | ✔ |
| Attribute-forecast | | ✔ | | ✔ | ✔ | ✔ |
| Extensible | | | | | ✔ | ✔ |
| Multi-task | | ✔ | | ✔ | ✔ | ✔ |
| Interpretable | | | | | | ✔ |

Table 1: Comparisons of NetEvolve with existing methods.

structure by extracting changes in the latent variables of nodes and the community structure. STEP [11], Dyrep [42], and VGRNN [20] are network structure prediction methods that incorporate structural and temporal information for link prediction. ONE-M [36] and TensorCast [4] are methods for predicting changes in node features from the network structure. These studies have focused only on predicting the graph structure or node's attributes, which can not capture the mutual effects between the change of the friendship and the change of their interests, a.k.a. attributes. In contrast to these studies, we focus on both predicting the graph structure and the attribute values of nodes that can fully utilize their mutual effects.

In a previous study, DualCast [27], a method for predicting both the future of a network structure and attributes, was proposed and showed high accuracy compared to existing methods. SINN [37] uses a mathematical model to describe properties known in psychology and incorporates them into the model to predict opinion dynamics. This research shows the effectiveness of describing the properties of psychology in a mathematical model for forecasting opinion dynamics; however, the generation of edges and the prediction of features for multiple future time series have not been sufficiently investigated.

Table 1 summarizes the characteristics of the proposed method compared to existing methods. In this study, we model properties known from network science and psychology and assume that nodes in a social network take the best actions to increase the degree of satisfaction. Compared to the existing methods, the proposed method differs significantly in considering changes in graph structure and attribute values, and allowing other properties to be easily incorporated.

## 3 PROBLEM DEFINITION

In this section, we explain the definition of the problem that we are targeting in this study. The input social network is represented as a graph structure with attribute values, where each node $n$ represents a person, the attribute value $\mathbf{x}_n \in \mathbb{R}^k$ is a vector representing the interests of each person such as words, and edges represent connections between people. Note that, for simplicity, we do not consider the appearance and disappearance of nodes. Appendix A.1 summarizes the symbols and their definitions.

We can define the objective of our research as follows:

- **Input**: Given social networks $\mathcal{G}^{(t)} = \langle \mathcal{V}^{(t)}, \mathcal{E}^{(t)}, \mathcal{X}^{(t)} \rangle$ at time $t = 1 \sim T$, where $\mathcal{V}^{(t)}$ is the set of nodes, $\mathcal{E}^{(t)} =$ $\{e_{i,j}^{(t)}\}$ is the set of edges between nodes, and $\mathcal{X}^{(t)} = \{\mathbf{x}_i\}$ is the set of attribute value vectors.
- **Output**: Forecast the future, that is, the of edges $\mathcal{E}^{(t')}$ and node-attributes $\mathcal{X}^{(t')}$ at $t' > T$.

## 4 PROPOSED METHOD: NETEVOLVE

This section describes NetEvolve, a model for predicting future social networks based on multi-agent reinforcement learning. Note that we construct the model that the parameters are interpretable to understand the property of the given social networks, and easy to extend by incorporating the other knowledge on network science.

The method consists of two stages; (1) optimize reward and policy functions for each node using historical time-series social network data, and (2) generate unobserved time-series network data based on multi-agent simulation.

The proposed method learns the parameters of each node's reward function from the input social network time series and optimizes the policy function to maximize the estimated value of the reward function using the policy gradient method. The future network is generated by calculating the probability of edge creation and deletion, and change of their attributes using the learned parameters of the policy functions.

### 4.1 Reinforcement Learning Environment

*4.1.1 Preliminary: Markov Decision Process.* In this study, we assume the environment as the Markov decision process (MDP)[6], which consists of the state $S^{(t)}$, action $A^{(t)}$, and reward $R^{(t)}$. In MDP, the agents select the actions based on the state by the policy function $\pi(A^{(t)} \mid S^{(t)}, \Theta)$, and the state will change according to the agent's actions according to the state transition function $f(S^{(t+1)} \mid S^{(t)}, A^{(t)})$, and the rewards are calculated based on the reward function $r(S^{(t)} \mid \Psi)$, where $\Theta$ and $\Psi$ are the parameters of the policy function and the reward function, respectively. The objective of reinforcement learning is to learn the parameter for the policy function that maximizes the expected reward.

*4.1.2 MDP for Social Network.* In this study, to utilize reinforcement learning to forecast the social network, we define

- **State** $S^{(t)}$ as the current social network $\mathcal{G}^{(t)}$.
- **Action** $A^{(t)}$ as the change in social network $\Delta \mathcal{G}^{(t)}$.

Note that, $\Delta \mathcal{G}^{(t)} = \langle \Delta \mathcal{E}^{(t)}, \Delta \mathcal{X}^{(t)} \rangle$, where $\Delta \mathcal{E}^{(t)}$ is a set of newly added/deleted edges and $\Delta \mathcal{X}^{(t)}$ is a change in attributes of nodes. By using the above statements, we can describe the reward, policy, and state transition as follows:

- **Reward** $r(S^{(t)} \mid \Psi) = r(\mathcal{G}^{(t)} \mid \Psi)$
- **Policy** $\pi(A^{(t)} \mid S^{(t)}, \Theta) = \pi(\Delta \mathcal{G}^{(t)} \mid \mathcal{G}^{(t)}, \Theta)$
- **State transition**
  $f(S^{(t+1)} \mid S^{(t)}, A^{(t)}) = \langle \mathcal{V}^{(t)}, \mathcal{E}^{(t)} \cup \Delta \mathcal{E}^{(t)}, \mathcal{X}^{(t)} \cup \Delta \mathcal{X}^{(t)} \rangle$

We can forecast the future social network by using the policy function $\pi(\Delta \mathcal{G}^{(t)} \mid \mathcal{G}^{(t)}, \Theta)$ and the state transition $\langle \mathcal{V}^{(t)}, \mathcal{E}^{(t)} \cup \Delta \mathcal{E}^{(t)}, \mathcal{X}^{(t)} \cup \Delta \mathcal{X}^{(t)} \rangle$.

*4.1.3 Multi-agent Reinforcement Learning for Social Network.* We assume that each node $n_i$ is an agent and has a reward function $r_i(\mathcal{G}^{(t)} \mid \psi_i)$ with a set of parameters $\psi_i \in \Psi$ and a policy function $\pi_i(\Delta \mathcal{G} \mid \mathcal{G}^{(t)}, \theta_i)$ with a set of parameters $\theta_i \in \Theta$; which represents

as the multi-agent environment. The reward function expresses each node's desirability to a social network, and the policy function expresses the tendency to change their edges and attributes. For simplicity, we assume the overall reward is a summation of each node's reward and the policy is the simple product of each node's policy, which are described as follows:

- **Reward** $r(\mathcal{G}^{(t)} \mid \Psi) = \sum_{n_i \in \mathcal{V}} r_i(\mathcal{G}^{(t)} \mid \psi_i)$
- **Policy** $\pi(\Delta \mathcal{G}^{(t)} \mid \Theta) = \prod_{n_i \in \mathcal{V}} \pi_i(\Delta \mathcal{G}^{(t)} \mid \mathcal{G}^{(t)}, \theta_i)$

We construct the optimization scheme to learn the parameters $\Psi = \{\psi_i\}_{n_i \in \mathcal{V}}$ of the reward function and $\Theta = \{\theta_i\}_{n_i \in \mathcal{V}}$ of the action policy from the sequence of social network $\langle \mathcal{G}^{(1)}, \mathcal{G}^{(2)}, \dots, \mathcal{G}^{(T)} \rangle$.

*4.1.4 Forecasting the Network using the Policy Function.* By learning the policy function, we can forecast the future sequence of the social network $\langle \mathcal{G}^{(T+1)}, \mathcal{G}^{(T+2)}, \dots, \mathcal{G}^{(T+T')} \rangle$. Based on MDP, the probability of generating a sequence of future social network is

$$Pr(\langle \mathcal{G}^{(T+1)}, \dots, \mathcal{G}^{(T+T')} \rangle) = \prod_{t=1}^{T'} \pi(\Delta \mathcal{G}^{(T+t)} \mid \mathcal{G}^{(T+t)}, \Theta) \quad (1)$$

## 4.2 Reward Function

In this section, we describe the design of the reward function for a given social network. We designed the reward function to measure the desirability of a given network for each node. In this work, we assume the reward that a node gets from the network is calculated based on the relationship between neighboring nodes (Fig. 3). We define the reward function in the social network at time $\mathcal{G}^{(t)}$ for each node by the following equation.

$$r_i(\mathcal{G}^{(t)} \mid \psi_i) = \sum_{\substack{n_j \in \mathbf{N}(n_i)}} \alpha_i\, sim(n_i, n_j) - \beta_i\, cost(n_i, n_j) + \gamma_i\, impact(n_j) \quad (2)$$

Eq. 2 consists of a linear combination of the reward based on the similarity of the connecting nodes, the cost of connecting an edge, and the impact for changing the neighbor's attribute with weighting parameters $\alpha_i, \beta_i$ and $\gamma_i$. Note that the reward function is easy to extend by adding the other factors for calculating the reward of the nodes. Intuitively, each of the parameters can be interpreted as follows:

- $\alpha_i$: **Similarity** weight representing tendency for making homophily. When this value is positive, the node is more motivated to connect to someone with close interests.
- $\beta_i$: **Cost** weight representing stress in human connections. The higher this value, the lower the reward value for a connection.
- $\gamma_i$: **Impact** weight representing reward for influencing neighbors. When this value is positive, the node gets a higher reward by influencing the neighbor's attribute.

where $\psi_i = \{\alpha_i, \beta_i, \gamma_i\}$ is the parameter set of node $n_i$ and $\mathbf{N}(n_i)$ is the set of adjacent nodes of $n_i$.

$sim(n_i, n_j)$ denotes the similarity of nodes. In this study, for simplicity, we assume the similarity as the cosine similarity of the node attribute value vector $\mathbf{x}_i$.

$$sim(n_i, n_j) = \frac{\boldsymbol{x}_i \cdot \boldsymbol{x}_j}{|\boldsymbol{x}_i||\boldsymbol{x}_j|} \quad (3)$$

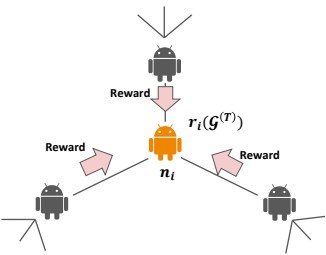

**Figure 3: Evaluation scheme for reward function. The reward is calculated for each edge based on similarity and the cost and aggregated over the neighboring nodes.**

$cost$ denotes the cost of having an edge between node $n_i$ and $n_j$ in the social network. In this study, we defined it as follows.

$$cost(n_i, n_j) = \begin{cases} 1, & e_{i,j} \in \mathcal{E}^{(t)} \\ 0, & e_{i,j} \notin \mathcal{E}^{(t)} \end{cases} \quad (4)$$

$impact$ denotes the influence of one's attributes on others. In this study, we defined it as follows.

$$impact(n_j) = \|\mathbf{x_j}^{(t)} - \mathbf{x_j}^{(t-1)}\|_2^2 \quad (5)$$

We can also employ more complex models for $sim$, $cost$, and $impact$, such as utilizing embeddings and graph neural networks [8, 13, 28].

To optimize the parameters in the reward function, we assume that the observed time-series social network is optimal in the environment and estimate the parameters so that the value of the reward function in the current social network is maximized. In this study, we find the parameter that maximizes the reward summed over a series of input social networks.

$$\Psi^* \leftarrow argmax_\Psi \sum_{t=1}^{T} r(\mathcal{G}^{(t)} \mid \Psi) \quad (6)$$

To optimize the problem, we employed SGD to estimate the parameters that maximize the reward.

## 4.3 Policy Function

This section describes the policy function of the nodes. We assume each node can take the actions for making/deleting edges and changing their attributes. In this study, the policy for edges and the attributes are independent,

$$\pi_i(\Delta \mathcal{G}^{(t)} \mid \mathcal{G}^{(t)}, \theta_i) = \pi_i(\Delta \mathcal{E}^{(t)} \mid \mathcal{G}^{(t)}, \theta_i) \cdot \pi_i(\Delta \mathcal{X}^{(t)} \mid \mathcal{G}^{(t)}, \theta_i) \quad (7)$$

In the following sections, we describe the design of the policy functions and strategy for learning the parameters.

*4.3.1 Policy Function for Edges.* In this section, we describe the design of policy functions for changing their edges. The choices of action of a node are making new edges or deleting the existing edges. We assume the targets to create edges based on random selection and transitivity based on knowledge of network science [45]. In this study, let the making/deleting of the edges be independent; we

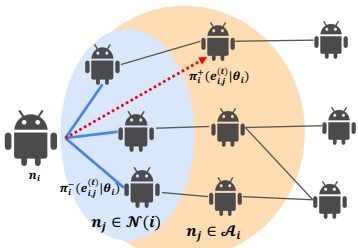

**Figure 4: How to run policy function for edges. The policy for making new edge $\pi^+(\cdot)$ calculates the edge probability for the two-hop neighbors and random nodes, and the policy for deleting edge $\pi^-(\cdot)$ calculates the deleting probability for the current edges.**

assume the policy function can be decomposed as follows:

$$\pi_i(\Delta \mathcal{E}^{(t)} \mid \mathcal{G}^{(t)}, \theta_i) = \prod_{n_j \in \mathcal{A}_i} \pi_i^+(e_{i,j}^{(t)} \mid \theta_i) \prod_{n_j \in N(n_i)} \pi_i^-(e_{i,j}^{(t)} \mid \theta_i) \tag{8}$$

where $\pi_i^+(e_{i,j} \mid \theta_i)$ and $\pi_i^-(e_{i,j} \mid \theta_i)$ are the policy functions for making and deleting an edge, respectively. Let $\mathcal{A}_i = \{n_j \in \mathcal{V} \mid n_i \in V_{neighbor}(i) \cup V_{random}\}$ as action space for making new edges, where $V_{neighbor}(i) = \bigcup_{n_j \in N(i)} N(i)$ which is a set of nodes in the two-hop neighbors of node $n_i$, and $V_{random}$ is a set of nodes which are randomly selected. By designing the action space as the above setting, the edges are more likely to be generated in the neighbor of the neighbors, which simulates the transitivity phenomenon.

We design the policy function for making/deleting the edges to illustrate the variety of representation patterns (Fig. 4). To increase the reward function, there are a variety of strategies, such as (1) making the edges too many people while ignoring the costs, (2) making the edges a limited number of people with similar attributes, etc. In this study, we define a similarity function based on the attribute values of each node and use it to define a policy function. Using the similarity function $sim(n_i, n_j)$, the policy function for edge making $\pi^+(\cdot)$ and edge deletion $\pi^-(\cdot)$ for the $i$-th node are defined as follows:

$$\pi_i^+(e_{i,j}^{(t)} \mid \theta_i) = \tanh\left\{\epsilon_i \exp\left(\frac{sim(n_i, n_j)}{\tau_i}\right)\right\} \tag{9}$$

$$\pi_i^-(e_{i,j}^{(t)} \mid \theta_i) = \tanh\left\{\epsilon_i \exp\left(\frac{1 - sim(n_i, n_j)}{\tau_i}\right)\right\} \tag{10}$$

Equations (9)(10) contains the following parameters:

- $\epsilon_i$: **Intensity** for making/deleting the edges. When the parameter is large, the node tends to make/delete more of the edges to increase the rewards.
- $\tau_i$: **Temperature** for making/deleting the edges. When the parameter is small, the node tends to make/delete edges relying on the similarity of the attributes.

*4.3.2 Policy Function for Attributes.* Next, this section describes the policy function for changing the attributes. The function is designed to illustrate various stories of changing the attribute to earn higher rewards (Fig.5). The possible stories are (1) changing their attributes similar to the neighboring nodes to increase the similarity between the neighbors, (2) enforcing the neighbors to

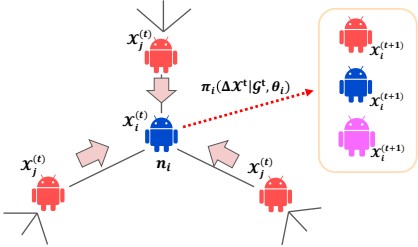

**Figure 5: How to run policy function for attributes. In this figure, the color of the node represents the attributes. The nodes change their attributes based on his/her stubbornness and the attributes of neighboring nodes of strong influence.**

change the attributes to make the neighboring nodes similar, etc. To illustrate such stories, we model the policy based on the nodes' stubbornness and influence. To simplify the function, we designed the policy function can make changes to their own attributes only:

$$\pi_i(\Delta \mathcal{X}^{(t)} \mid \mathcal{G}^{(t)}, \theta_i) = \pi_i(x_i^{(t+1)} \mid \mathcal{G}^{(t)}, \theta_i)$$
$$= \sigma\left(\lambda_i \cdot x_i^{(t)} + (1 - \lambda_i) \sum_{n_j \in N(n_i)} \mu_j \cdot x_j^{(t)}\right) \tag{11}$$

Where $\lambda_i, \mu_i$ are the parameter of node $i$.

- $\lambda_i$: **Stubbornness** of their attributes. When the value is large means the node does not tend to change its attributes.
- $\mu_i$: **Influence** to the neighbors. The larger value means they influence to make change the attribute of neighbors.

Overall, the set of parameters in policy functions of node $n_i$ is $\theta_i = \{\epsilon_i, \tau_i, \lambda_i, \mu_i\}$, i.e., the behavior of the node $n_i$ in the network is characterized by the parameters.

*4.3.3 Learning the Parameters in Policy Functions.* Next, we describe how to learn the parameters of the policy functions. Specifically, we learn the parameters of the policy function that maximizes its own optimized reward function in 4.2 using REINFORCE [47], one of the gradient descent methods. This method generates an unobserved social network and learns the policy parameters that maximize the expected value of the reward function.

Assuming that each node in the social network has chosen the optimal action at each time, it is possible to predict the network in the unobserved time series by obtaining the strategy that maximizes the future reward.

In this study, we optimize the policy function parameter $\Theta$ using the policy gradient method. To generate a social network at an unobserved time based on the policy function $\pi(\Delta \mathcal{G}^{(t)} \mid \mathcal{G}^{(t)}, \Theta)$, we use the generation function eq. 1, and we note the simulated sequence as $\omega = \langle \mathcal{G}^{(T+1)}, \mathcal{G}^{(T+2)}, \ldots, \mathcal{G}^{(T+T')} \rangle$. From the sequence, we can calculate the cumulative rewards for all nodes in the network at an unobserved time as follows.

$$R(\omega) = r(\mathcal{G}^{(T+1)}) + \xi \cdot r(\mathcal{G}^{(T+2)}) + \cdots + \xi^{T'} \cdot r(\mathcal{G}^{(T+T')}) \tag{12}$$

where $\xi$ is the discount rate.

This method learns the policy function's parameters that maximize the cumulative reward's expected value. The objective function $J(\theta)$ is expressed as follows.

$$J(\Theta) = \mathbb{E}_{\omega \sim \pi_\Theta}[R(\omega)] \tag{13}$$

We derive the gradient for the parameter $\Theta$ in the objective function to maximize the above objective function.

$$\nabla_\Theta J(\Theta) = \nabla_\Theta \mathbb{E}_{\omega \sim \pi_\Theta}[R(\omega)]$$

$$\approx \frac{1}{N} \sum_{i=1}^{N} \sum_{t=1}^{T'} R(\omega^{(i)}) \nabla_\Theta \log \pi(\Delta \mathcal{G}^{(t)} \mid \mathcal{G}^{(t)}, \Theta) \tag{14}$$

Refer to appendix A.2 to see the derivation process. $\omega^{(i)}$ is the $i$-th sequence sampled by the policy function. The gradient is used to update the set of parameters $\Theta$

$$\Theta^* \leftarrow \Theta + \eta \cdot \nabla_\Theta J(\Theta) \tag{15}$$

where $\eta$ is the learning rate. The learned parameters of the policy functions are used to generate an unobserved network graph. Appendix A.3 shows the computational complexity of NETEVOLVE is almost linear to the number of nodes in a sparse network.

## 5 EXPERIMENT 1: ADEQUACY OF MODEL

To answer (RQ1), we conducted experiments with synthesized data to see whether NETEVOLVE can simulate potential phenomena we can see in social networks, such as homophily, heterophily, homogenization, and polarization [26, 33, 34, 39]. We expect that the parameters of NETEVOLVE can explain such phenomena with a variety of parameter settings.

### 5.1 Experimental Setting

In this experiment, we manually set the parameters of the reward function and the policy function, and see in what setting the phenomena will occur. The phenomena we aim to simulate are *homophily*, *heterophily*, *homogenization*, and *polarization*, which are very famous phenomena in real-world social networks. The following are brief explanations of each pattern.

- **Homophily**: The property of the similar nodes in the network tends to be connected in the network. In a homophily network, the network will grow by people connecting to similar people [9, 24].
- **Heterophily**: The property of the dissimilar nodes in the network tends to be connected in the network. In a heterophily network, the network will grow by people connecting dissimilar nodes [49].
- **Homogenization**: The property of the connecting nodes gradually gets similar to each other while keeping the connections [39].
- **Polarization**: The property of the similar nodes form communities, and the communities are gradually separated. The phenomena typically observed in the congress network [15, 18, 22, 26].

In this experiment, we set the number of nodes is 20, and the attributes are "red" and "blue".

## 5.2 Results: Representation Differences in Parameters

Figures 6a to 6d show the generated network, and table 2 summarizes the parameter settings to simulate the networks. In figure 6a, NETEVOLVE generates *homophily* network, in which the parameters are a high reward in high similarity, and the low cost for connection, and the nodes have great influence and low stubbornness. This illustrates the nodes will be happy to connect and make one big community of similar nodes. In figure 6c, NETEVOLVE generates *heterophily* network, in which the parameters are similar to the case of *homophily* in rewards and low influence. The final network is different from *homophily* network; one big community of nodes having different attributes occurs. In figure 6b, NETEVOLVE generates *homogenization*, in which the parameters are similar to *homophily*. In figure 6d, NETEVOLVE generates *polarization*, the community is separated, and the nodes of the same attribute form the different communities. These results indicate that, even if the initial network is in the same condition, NETEVOLVE can successfully illustrate the different scenarios in different parameters.

## 6 EXPERIMENT2: FIT NETEVOLVE AND FORECAST IN REAL-WORLD DATA

To answer (RQ2), we verify how accurately NETEVOLVE predicts unobserved social network edges and changes in attributes using real data. More specifically, we calculated the probability that the edges and the attribute were generated in the unknown time segment. The details of implementation and the hyperparameter setting are shown in appendix A.5.

As for comparative methods, for edge forecasting, we employ simple RNN, DualCast [27], which predicts edges and features based on latent vector features, and VGAE [29] and VGRNN [20], which are methods based on graph neural networks that generate edges. For attribute forecasting, we employ simple RNN and DualCast. Moreover, for the ablation study, we employed NETEVOLVE with only forecasts edge or attributes.

In this study, we employ the area under the curve (AUC) and negative log-likelihood (NLL) to verify the forecasting accuracy, where a higher AUC and lower NLL indicate a more accurate prediction.

### 6.1 Dataset Description

We use the three datasets, `DBLP`, `NIPS`, and `Twitter` for experiments. We chose them to cover a variety of settings in terms of size and density. In every dataset, we collected 10 time segments and used 5 time segments to optimize the reward function and the remaining to test the accuracy. The following are the dataset description and statistics (see appendix A.4 for more detail):

- `DBLP` is a co-authorship network of researchers. Nodes are authors, edges are co-authorship, and the time segment is in years. The number of nodes is 32 and the average *density* of edges is $0.0045 \pm 0.00020$, where $density \triangleq |\mathcal{E}|/_{|\mathcal{V}|}C_2$.
- `NIPS` is a co-authorship network of researchers. Nodes are authors, edges are co-authorship, and the time segment is in years. The number of nodes is 500 and the average *density* of edges is $0.0435 \pm 0.00259$.

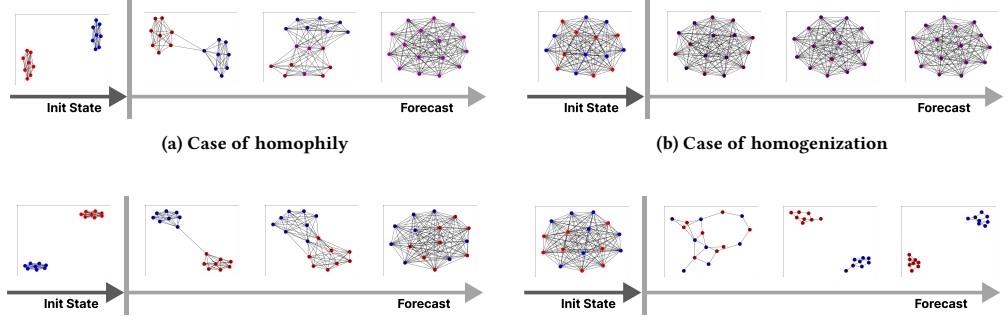

(a) Case of homophily              (b) Case of homogenization

(c) Case of heterophily              (d) Case of polarization

**Figure 6: Results of EXPERIMENT1: By setting the different parameters, NETEVOLVE can simulate the various types of social network phenomena. In each scenario, the parameters are set to that stated in table 2.**

| | $\alpha$ (similarity) | $\beta$ (cost) | $\gamma$ (impact) | $\epsilon$ (intensity) | $\tau$ (temperature) | $\lambda$ (stubbornness) | $\mu$ (influence) |
|---|---|---|---|---|---|---|---|
| **homophily** (Fig.6a) | *Very High* | *Low* | *High* | *High* | *High* | *High* | *Very High* |
| **homogenization** (Fig.6b) | *High* | *Low* | *High* | *Low* | *High* | *Low* | *Very High* |
| **heterophily** (Fig.6c) | *Very High* | *Low* | *Low* | *High* | *High* | *High* | *Low* |
| **polarization** (Fig.6d) | *High* | *High* | *Low* | *High* | *Low* | *High* | *Low* |

**Table 2: Parameter setting of EXPERIMENT1. The $(Very High, High, Low)$ represents the relative value of parameters. We note "$Very High$" if $x \geq 1.5$, "$High$" if $1.5 > x \geq 1.0$, and "$Low$" if $x < 1.0$. For simplicity, we set the parameters of every node to the same.**

- Twitter is a retweet network of Twitter users, and the time segment is in months. The number of nodes is 15000 and the average *density* of edges is $2.79e - 6 \pm 1.83e - 6$.

We use the NIPS and DBLP, the first five years are used for training data, and the remaining five years are used for test data. And, we use the twitter datasets, the first five months are used for training data, and the remaining data is used for testing.

## 6.2 Result: Accuracy Evaluation on Forecasting

Figures 7a to 7c show the change in AUC values for edge forecasting, and figures 7d to 7f show the NLL at each prediction time. By comparing the accuracy of NETEVOLVE and that of predicting only edges, the two methods achieve almost the same accuracy in DBLP and NIPS. In Twitter, NETEVOLVE of both predicting the edges and attributes is higher than that for only edges. It indicates that multi-task learning is effective in several cases. Figures 8a to 8c show the change in AUC values for attribute forecasting, and figures 7d to 7f show the NLL at each prediction time. Similar to the edge forecasting task, NETEVOLVE forecasting edges and attributes wins that of only forecasting edges. The running time for the experiment were $49.6 \pm 1.31(s)$, $10.0 \pm 3.16(s)$, and $1213.9 \pm 199.0(s)$ for DBLP, NIPS, and Twitter, respectively. It indicates our method is scalable for large network data having over 10,000 nodes.

## 6.3 Interpretability: Learned Parameters

To validate the interpretability of NETEVOLVE, we monitored the optimized parameters of reward functions. Table 3 shows the average and std of learned parameters $\alpha$ (similarity), $\beta$ (cost), and $\gamma$ (impact). Before learning the parameters, we set the initial value of each parameter to 1.0. The results of DBLP, NIPS, and Twitter show that $\alpha$ (similarity) and $\gamma$ (impact) have a larger impact on node behavior than $\beta$ (cost). A result on the Twitter shows the

| Dataset | | average ± std. |
|---|---|---|
| DBLP | $\alpha$ (similarity) | $1.31 \pm 0.40$ ($High$) |
| | $\beta$ (cost) | $0.93 \pm 0.02$ ($Low$) |
| | $\gamma$ (impact) | $3.00 \pm 0.00$ ($Very High$) |
| NIPS | $\alpha$ (similarity) | $1.13 \pm 0.12$ ($High$) |
| | $\beta$ (cost) | $0.99 \pm 0.00$ ($Low$) |
| | $\gamma$ (impact) | $1.13 \pm 0.00$ ($High$) |
| Twitter | $\alpha$ (similarity) | $1.03 \pm 1.29$ ($High$) |
| | $\beta$ (cost) | $0.99 \pm 0.00$ ($Low$) |
| | $\gamma$ (impact) | $41.0 \pm 0.00$ ($Very High$) |

**Table 3: The result of learned parameters in each dataset.**

effects of $\alpha$ and $\beta$ are the same and $\gamma$ has a larger impact on node behavior.

## 7 CONCLUSION

We proposed a novel multi-agent reinforcement learning-based network forecasting method called NETEVOLVE, that explicitly models the network science and psychology knowledge by designing a reward function and policy function, which makes the model explainable and can derive network scientific outcome from the model parameters. Experiments show NETEVOLVE can simulate the various types of social phenomena and can forecast future networks comparably or more accurately than the related works, which indicates NETEVOLVE well fits the change in real-world social networks.

For future works, we aim to explain the behavior of nodes in social networks with higher accuracy by incorporating properties revealed in other network sciences and psychology into the reward function that determines behavior.

**Reproducability:** Our code and datasets will be open-sourced on GitHub https://github.com/crowd4u/netevolve.

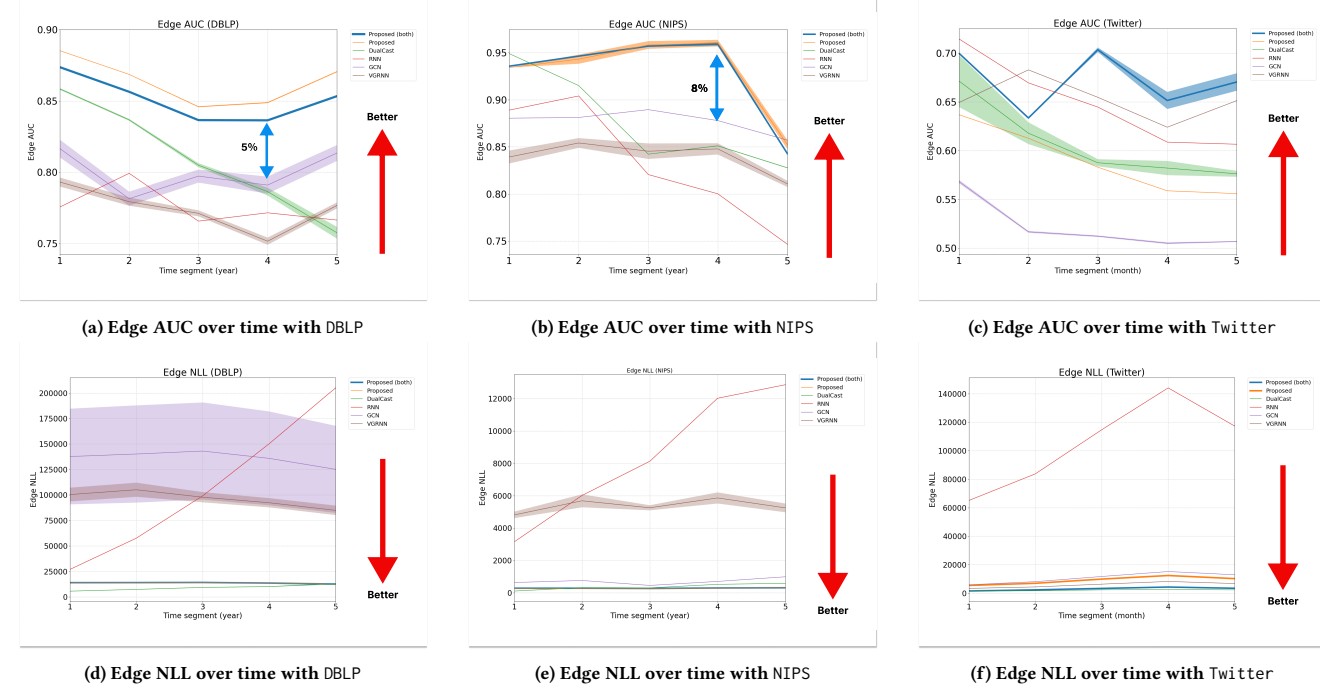

**(a) Edge AUC over time with** DBLP          **(b) Edge AUC over time with** NIPS          **(c) Edge AUC over time with** Twitter

**(d) Edge NLL over time with** DBLP          **(e) Edge NLL over time with** NIPS          **(f) Edge NLL over time with** Twitter

**Figure 7: The accuracy of edge forecasting: These figures show the AUC and NLL of forecasting the edges that appear in the future. "Proposed(both)" is NetEvolve and "proposed" is a method that only considers the policy function for edges. Both methods win the comparative methods in most cases.**

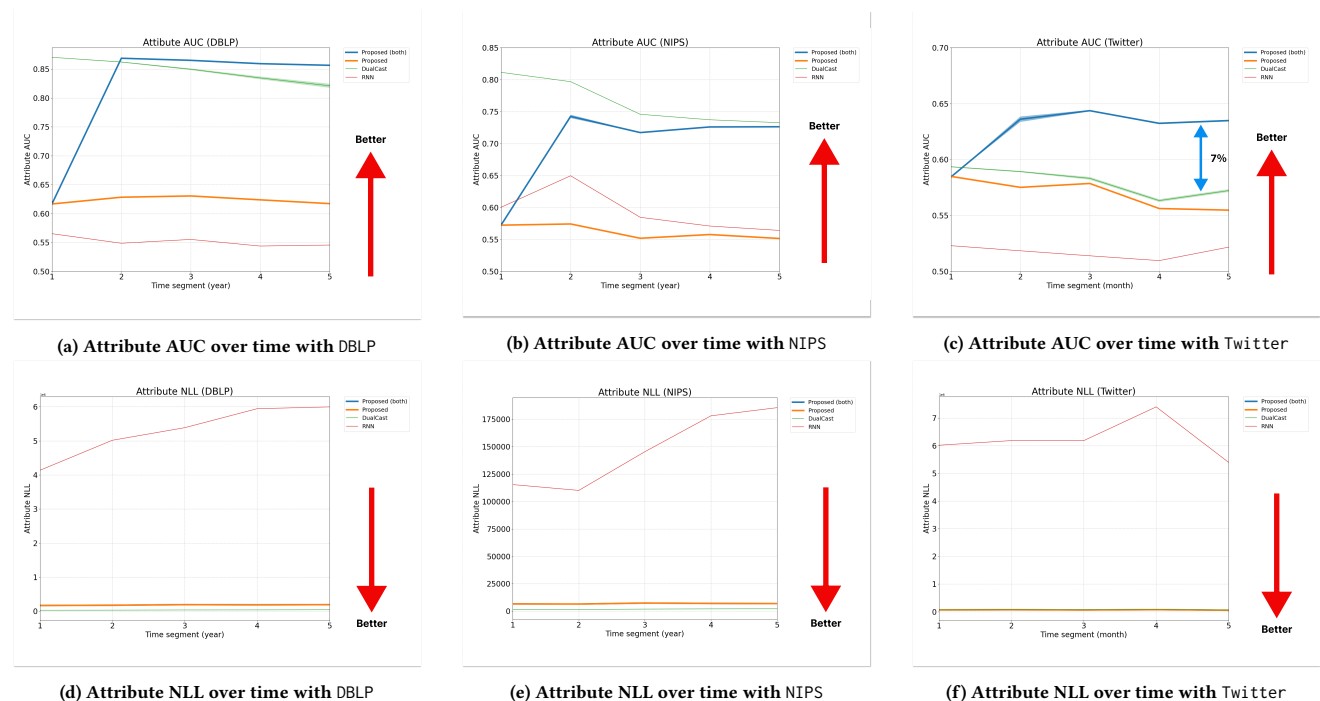

**(a) Attribute AUC over time with** DBLP          **(b) Attribute AUC over time with** NIPS          **(c) Attribute AUC over time with** Twitter

**(d) Attribute NLL over time with** DBLP          **(e) Attribute NLL over time with** NIPS          **(f) Attribute NLL over time with** Twitter

**Figure 8: The accuracy of attribute forecasting: These figures show the AUC and NLL of forecasting the attribute changes in the future. "Proposed(both)" is NetEvolve and "proposed" is a method that only considers the policy function for attributes. NetEvolve win the comparative methods in most of the cases.**

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

# A APPENDICES

## A.1 Table of symbols

For referencing the symbols, we state the table as a summary.

### Table 4: Symbols and Definitions

| | |
|---|---|
| $\mathcal{G}^{(t)}$ | Graph at time $t$. |
| $\mathcal{V}^{(t)} = \{n_i\}$ | Node set at time $t$. |
| $\mathcal{E}^{(t)} = \{e_{i,j}^{(t)}\}$ | Edge set at time $t$. |
| $\mathcal{X}^{(t)} = \{x_i\}$ | Attribute value set at time $t$. |
| $n_i$ | $i$-th node. |
| $e_{i,j}^{(t)} = \langle n_i, n_j \rangle$ | Directed edge between nodes $n_i$ and $n_j$. |
| $x_i \in \mathbb{R}^k$ | Attribute of node $n_i$. |
| $S^{(t)}$ | State at time $t$. |
| $A^{(t)}$ | Action at time $t$. |
| $r(S^{(t)} \mid \Psi)$ | Reward function with parameters $\Psi$. |
| $\pi(A^{(t)} \mid S^{(t)}, \Theta)$ | Policy function with parameters $\Theta$. |
| $\Psi = \{\psi_i\}_{n_i \in \mathcal{V}^{(t)}}$ | Parameters of reward function. |
| $\Theta = \{\theta_i\}_{n_i \in \mathcal{V}^{(t)}}$ | Parameters of policy function. |
| $\psi_i = \{\alpha_i, \beta_i, \gamma_i\}$ | Parameters of reward function of node $n_i$. |
| $\theta_i = \{\epsilon_i, \tau_i, \lambda_i, \mu_i\}$ | Parameters of policy function of node $n_i$. |
| $\alpha_i \in \mathbb{R}^+$ | **Similarity** weight in reward. |
| $\beta_i \in \mathbb{R}^+$ | **Cost** weight in reward. |
| $\gamma_i \in \mathbb{R}^+$ | **Impact** weight in reward. |
| $\epsilon_i \in \mathbb{R}^+$ | **Intensity** in policy for edges. |
| $\tau_i \in \mathbb{R}^+$ | **Temperature** in policy for edges. |
| $\lambda_i \in \mathbb{R}^+$ | **Stubbornness** in policy for attributes. |
| $\mu_i \in \mathbb{R}^+$ | **Influence** in function for attributes. |
| $\eta \in \mathbb{R}^+$ | Learning rate of policy gradient. |
| $\xi \in \mathbb{R}^+$ | Discount rate of the reward. |
| $N \in \mathbb{N}^+$ | Number of simulations in policy gradient. |
| $T' \in \mathbb{N}^+$ | Future time of the simulation. height |

## A.2 Derivation of the policy gradient

We derive the gradient for the parameter $\Theta$ in the objective function (13) to maximize the function. Following the "log-likelihood trick" [46], we derive the gradient:

$$
\begin{aligned}
\nabla_\Theta J(\Theta) &= \nabla_\Theta \mathbb{E}_{\omega \sim \pi_\Theta} [R(\omega)] \\
&= \nabla_\Theta \sum_\omega Pr(\omega \mid \Theta) R(\omega) \\
&= \sum_\omega R(\omega) \nabla_\Theta Pr(\omega \mid \Theta) \\
&= \sum_\omega R(\omega) Pr(\omega \mid \Theta) \frac{\nabla_\Theta Pr(\omega \mid \Theta)}{Pr(\omega \mid \Theta)} \\
&= \mathbb{E} \left[ R(\omega) \nabla_\Theta \log Pr(\omega \mid \Theta) \right] \\
&= \mathbb{E} \left[ \sum_{t=1}^{T'} R(\omega) \nabla_\Theta \log \pi(\Delta \mathcal{G}^{(t)} \mid \mathcal{G}^{(t)}, \Theta) \right] \\
&\approx \frac{1}{N} \sum_{i=1}^{N} \sum_{t=1}^{T'} R(\omega^{(i)}) \nabla_\Theta \log \pi(\Delta \mathcal{G}^{(t)} \mid \mathcal{G}^{(t)}, \Theta) \quad (16)
\end{aligned}
$$

## A.3 Computational Complexity

NETEVOLVE has the following part of the calculation each of which has the following time complexity :

(1) Learning reward function: $O(T \cdot |\mathcal{E}|)$

(2) Learning policy function: $O(T \cdot N \cdot \frac{|\mathcal{E}|^2}{|\mathcal{V}|})$

(3) Generate future networks: $O(T' \cdot \frac{|\mathcal{E}|^2}{|\mathcal{V}|})$

Overall, NETEVOLVE has $O\left((TN + T') \cdot \frac{|\mathcal{E}|^2}{|\mathcal{V}|}\right)$. Note that, when edges are sparse, the time complexity is almost linear to $|\mathcal{V}|$.

## A.4 Dataset conditions

Following are more details of the dataset creation processes.

- DBLP is a co-authorship network of researchers. Nodes are authors, edges are co-authorship relationships, and the time segment is in years. When two authors, x and y, publish a joint paper in a given year, a bi-directional directed edge is created between them. We obtained 47 international conference papers published from 2008 to 2017 in data mining, databases, natural language processing, machine learning, artificial intelligence, information retrieval, and computer vision. We use the data from 2008 to 2012 as training data and from 2013 to 2017 as test data. The number of nodes is 500, and the number of attribute values is 3854.
- NIPS is a co-authorship network of researchers. Nodes are authors, and edges are co-authors. Network data, such as edge information and features, are generated similarly to DBLP. Each node has a word in the title of a paper published in a certain year as an attribute value. The number of nodes is 32, and the number of attribute values is 2411.
- Twitter is a retweet network of Twitter users. A node represents a user. We set the time segment to be monthly. When a user $i$ retweeted a tweet from another user $j$ during the month, directed edge appeared from $i$ to $j$. $112,044$ users were collected [7]. The collection period of tweets was from January 2010 to October 2010. Data from January to May were used as training data, and from June to October were used as test data. Users who appeared only once throughout all time segments were excluded from the dataset. The number of attribute values was 5372. Each user had hashtags included in the tweets from each time division as the attribute value.

## A.5 Details of implementation and hyperparameters

We can set the hyperparameters by measuring the accuracy of validation data, which are separated from the observed time series graph. In the experiments, we used 3 time segments as training data, and the 2 remaining time segments as validation data, and used Optuna [2] to search the optimal hyperparameters. As a result of tuning, we got $\eta = 0.00015$, $\xi = 0.420$ for NIPS, $\eta = 0.079$, $\xi = 0.164$ for DBLP, $\eta = 0.00086$, $\xi = 0.725$ for Twitter, and $|\mathcal{A}_{random}| = 0.001 \times |\mathcal{V}|$, $N = 48$, $T' = 48$ for all dataset.

We implemented NETEVOLVE using PyTorch2.1.0. We ran the experiments on Mac OS 13.4.1, Apple M1 Max, 64GB RAM, Python3.11.4.

