# OpenReview forum: "NETEVOLVE: Social Network Forecasting using Multi-Agent Reinforcement Learning with Interpretable Features"
_ACM.org/TheWebConf/2024/Conference — TheWebConf24_

### Official Review · Reviewer_ycVR · 2023-11-09

**Novelty:** 6
**Technical Quality:** 6

**Review:**

Overview：
This paper claims that the change in a social network can be explained by a small number of concepts, such as “homophily” and “transitivity”. Current methods lack exploring in such aspects. This paper proposed a novel multi-agent reinforcement learning-based method, NetEvolve,  to simulate how network changes and make predictions.  This paper claims that NetEvolve has the properties of Network-forecast, Attribute-forecast, Extensible, Multi-task, and Interpretable.

strength：
1. This paper proposed a novel method, NetEvolve, by using multi-agent reinforcement learning to simulate real-world social networks. In a social network, each node represents one user and in this paper, it uses an agent to represent the node.
The reward function contains similarity, cost, and impact, which are consistent with real-world networks.
2.  This paper is well-organized and clearly written. The figures and tables are
informative. The problem and evaluation are clearly defined.
3. This paper included an anonymized Github link to the source code. Therefore, I believe this submission contains sufficient information that helps in reproducibility.


weakness：
1.  The cost of reward is not so reasonable, which is only related to degree. In real-world networks, users can link with many other users without much cost. For example, on Twitter, one user follows another.
2. In Eq. 2, the total reward of node i  contains the cost of impact  $n_j$, which is Eq. 5. However, Eq. 5 represents the difference of node j between timestep t and t-1. It is confusing and different from traditional MP-GNNs.
3. There is no hyperparameter analysis in this paper. In reinforcement learning, the learning parameters should be $\theta$ in Eq. 7 as far as I know. However, in this paper, each node's reward contains three learning parameters. It is not clear how to optimize these parameters.

**Questions:**

1. How is the time-consuming in the training stage?

**Ethics Review Description:**

No ethics issues in this paper

**Reviewer Confidence:**

3: The reviewer is confident but not certain that the evaluation is correct

**Scope:**

3: The work is somewhat relevant to the Web and to the track, and is of narrow interest to a sub-community

---

### Official Review · Reviewer_s4p6 · 2023-11-13

**Novelty:** 4
**Technical Quality:** 4

**Review:**

The paper presents an interpretable new method for predicting changes in social networks in the future.

Pros:
1. The paper addresses the important problem of predicting changes in social networks, which has practical applications in various fields.
2. The proposed method, NetEvolve, utilizes multi-agent reinforcement learning with interpretable features, allowing for a better understanding of the prediction results.
3. The paper incorporates concepts from network science and psychology, making it more comprehensive in its approach.

Cons:
1. There are numerous typos in the paper. For instance, in lines 486-487, the definition of $A_i$ may contain some typos. Furthermore, there are grammar mistakes in the paper, as seen in line 636.
2. The competitors in the experiments are relatively weak; only DualCast is relatively new. If this is a practical and well-studied problem, perhaps the authors can find more state-of-the-art methods for comparison?
3. The explanations about the coefficients in the reward function in the experimental results are not clear. For instance, in Sec. 6.3, what information do larger $\alpha$ and $\gamma$ convey?

**Questions:**

Questions:
1. Can you provide more insights into the parameters of the reward function and how they relate to concepts like homophily, heterophily, etc.?
2. In Table 2, why does generating heterophily graphs require a large $\alpha$ value? This is counterintuitive.
3. See Con 3.

**Ethics Review Description:**

Not applicable.

**Reviewer Confidence:**

3: The reviewer is confident but not certain that the evaluation is correct

**Scope:**

4: The work is relevant to the Web and to the track, and is of broad interest to the community

---

### Official Review · Reviewer_f948 · 2023-11-23

**Novelty:** 4
**Technical Quality:** 4

**Review:**

(This paper was subreviewed)

## Summary

Considering the numerous applications involving dynamics in relational data, the authors propose a novel reinforcement learning-based approach, NetEvolve, for social network change prediction. In their work, they have designed a model capable of forecasting both network structure and attributes while emphasising the efficiency and interpretability of their approach as key features. To achieve this, the authors developed a custom reward function and implemented different policies to model real-world behaviours. The authors evaluated their approach using both synthetic and real-world data. They demonstrated how NetEvolve could retrieve typical graph characteristics by varying their model parameters to produce various types of social graphs. Additionally, they illustrated how their model outperformed several existing frameworks in a temporal link and attribute prediction task.

## Strengths

S1: The work addresses interesting questions at the intersection of social network analysis and interpretability. It emphasises the lack of interpretability in existing literature approaches, particularly those based on deep learning, and proposes to tackle this challenge.

S2: The introduction of reinforcement learning agents aims to adapt and learn dynamic representations in social networks, with the design of different policies to capture real-world behaviours.

## Weaknesses

W1:  Some of the claims lack justifications and are not clearly discussed. For instance in l.105-107, authors evokes how Graph Neural Network based methods are lacking the use of transitivity whereas they rely on it in their proposed approach. This suggests authors are addressing a limit existing in the literature, yet this limit is not clear: why transitivity helps in predicting changes in dynamic networks? Why authors think it is necessary to rely on it?
Author's justifications for "interpretable features" are summarised by opposing them to latent feature-based approaches, the latter giving results from which authors argue it is "hard to derive scientific knowledge". However, it is unclear which knowledge can/should be derived in this context. If interpretability of the results is considered a major contribution of the work, it could be interesting to further develop justifications towards it.

W2: Positioning within the existing literature on the subject can be further detailed; authors mention the existing link between their work and psychology knowledge based approaches. It could be interesting to discuss the links between the proposed work and the DeGroot-Friedkin-based models which are designed to study opinion diffusion in social networks and are referred to as interpretable by prior work cited by authors [1].

W3: The overall clarity of the paper could be improved. For instance, some concepts, e.g. homophily and transitivity are used within the abstract and introduction, but not defined or explained and presuppose reader's knowledge about them (homophily is first described in l.161-162).  Interpretability is often referred to as a major requirement, but it remains difficult to understand the notion in some specific contexts, e.g. l299. Moreover, neural network based methods are criticised for their lack of interpretability, yet authors mention the possibility of using them to learn their parameters "sim", "cost" and "impact", which seems orthogonal with the main strength of the interpretable method. The contributions l.112-137 suffer the same issue; "agent in reinforcement learning setting" is mentioned without further explanation and the interpretable feature design is justified considering "knowledge of network science" , without any further reference. In a similar way, the reward function includes a "cost" which is described as the "stress in human connections". However, the intuition for this notion is not clear and may requires additional justification or references. Finally, some formulations sometimes make the understanding difficult, e.g. in l.86-88, l.298-301, l.496-498.

Note: It is nice that the authors provide code and data for reproducibility. Nevertheless, I would advise the author who prepared the git repository to avoid committing with an open profile, which breaks the anonymity.

[1] "Predicting Opinion Dynamics via Sociologically-Informed Neural Networks", Okawa et al., 2022

**Questions:**

I reckon that additional justifications to the principal claims (e.g. need for interpretability) could greatly improve the global understanding. A broader analysis of the literature would also help positioning the contributions in its context. Finally, several sentences could benefit from rewriting in order to deliver ideas more clearly.

**Ethics Review Description:**

--

**Reviewer Confidence:**

3: The reviewer is confident but not certain that the evaluation is correct

**Scope:**

4: The work is relevant to the Web and to the track, and is of broad interest to the community

---

### Official Review · Reviewer_iyfg · 2023-11-24

**Novelty:** 5
**Technical Quality:** 3

**Review:**

Pros:
1. A fascinating perspective that expanded my horizons. The article is well-structured, and the illustration design is quite reasonable, with good quality.
2. This paper proposes an interpretable and efficient edge prediction solution, and the experimental results are also promising

Cons:
1. In fact, the mechanism of Graph Neural Networks (GNNs) inherently assumes transitivity, as I don't believe the authors consider this to be a significant gap overlooked by most GNN algorithms.
1. The code has not been publicly released. (unsolved)

**Questions:**

No further comments.

**Ethics Review Description:**

No ethical issuses

**Reviewer Confidence:**

2: The reviewer is willing to defend the evaluation, but it is likely that the reviewer did not understand parts of the paper

**Scope:**

3: The work is somewhat relevant to the Web and to the track, and is of narrow interest to a sub-community

---

### Official Review · Reviewer_Y5yj · 2023-11-27

**Novelty:** 4
**Technical Quality:** 5

**Review:**

The paper proposes a time varying graph edge prediction framework based on reinforcement learning framework. The framework is mainly a MDP with all information of the graph (i.e. node, edge, features) taken into account. The predictions of edges and attributes are independent outputs. Three datasets and four baselines (mostly deep learning models for graphs) are compared.

Strengths:
+ The framework is straightforward but outperforms the baselines
+ The problem is well described.
+ I couldn't find any major flaws.

Weakness:
- The framework seems to be applicable for any time varying graphs that I somehow find it too generic. It seems like we just need to put all the attributes and nodes and edges then it could predict these properties well. Can we have some more insights?
- Since one of the claim is interpretability, can the paper explain some interpretations of the models? e.g. what are the important features that produce the predictions?

**Questions:**

Same for those weakness mentioned above.

**Reviewer Confidence:**

2: The reviewer is willing to defend the evaluation, but it is likely that the reviewer did not understand parts of the paper

**Scope:**

3: The work is somewhat relevant to the Web and to the track, and is of narrow interest to a sub-community

---

### Decision · Program_Chairs · 2024-01-22

**Decision:**

Accept

**Comment:**

The paper addresses an interesting topic of network evolution, and the paper's emphasis on the need for interpretable models in social network analysis is significant, as understanding and explaining the behavior of social networks are essential for various applications, including information dissemination, recommendation systems, and fraud detection. The originality of the paper lies in its use of reinforcement learning and network science to address dynamic network changes.

 The quality of the work is generally good, with a well-structured paper. The authors employ a multi-agent reinforcement learning framework to address the dynamic nature of social networks, and they emphasize interpretability as a key feature of their approach.

 Reviewers noted several weaknesses in the paper, including the need for clearer justifications for certain claims, such as the reliance on transitivity in the proposed approach, the lack of detailed discussions about the knowledge derived from interpretability, and the clarity of concepts and formulations in the paper. They also raised concerns about the reasonableness of the cost function in the reward structure, confusion related to the impact parameter, and the absence of a hyperparameter analysis. In response, the authors provided clarifications and justifications for their claims, particularly regarding the role of transitivity in social networks and the interpretability of their approach. They also promised to improve the clarity of writing. Additionally, the authors acknowledged the importance of a hyperparameter analysis and committed to including it in the revised paper to provide a more comprehensive view of their model's behavior and parameter tuning strategies.